# CHAIN OF IMAGES FOR INTUITIVELY REASONING

## ABSTRACT

The human brain is naturally equipped to comprehend and interpret visual information rapidly. When confronted with complex problems or concepts, we use flowcharts, sketches, and diagrams to aid our thought process. Leveraging this inherent ability can significantly enhance logical reasoning. However, current Large Language Models (LLMs) do not utilize such visual intuition to help their thinking. Even the most advanced vision language models (e.g., GPT-4V and LLaVA) merely align images into the textual space, which means their reasoning processes remain purely verbal. To mitigate such limitations, we present a Chain of Images (CoI) approach, which can convert complex language reasoning problems to simple pattern recognition by generating a series of images as intermediate representations. Furthermore, we have developed a CoI evaluation dataset encompassing 17 distinct domains where images can intuitively aid problem-solving. Based on this dataset, we aim to construct a benchmark to assess the capability of future multimodal LLMs to leverage images for reasoning. In supporting our CoI reasoning, we introduce a symbolic multimodal large language model (SyMLLM) that generates images strictly based on language instructions and accepts both text and image as input. Experiments on Geometry, Chess, Topological, Temporal and Common Sense tasks sourced from the CoI evaluation dataset show that CoI improves performance significantly over the pure-language Chain of Thoughts (CoT) baselines.

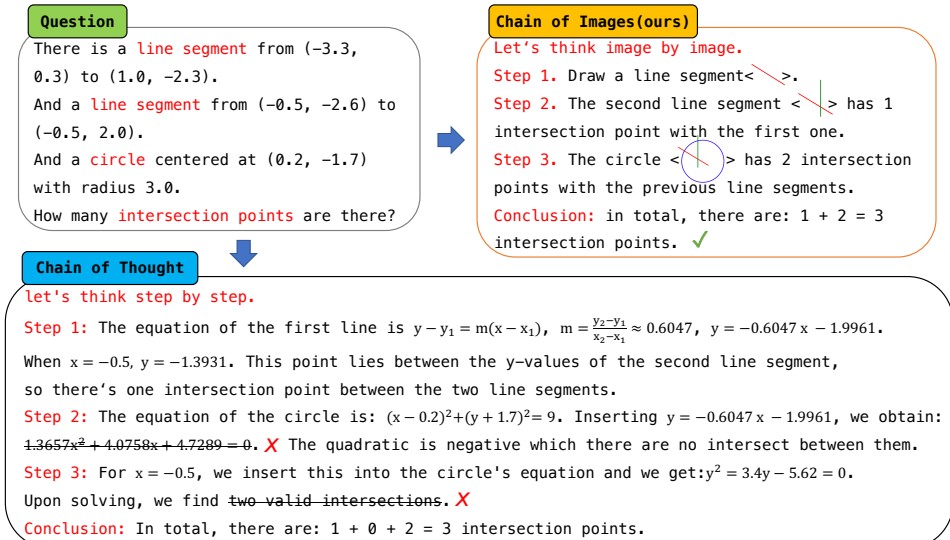

Figure 1: The simplified steps employing GPT-4 to calculate the number of intersection points using CoI are displayed in the bottom block. The whole processes of CoI are displayed in the top right block. Two issues were identified in CoT: 1) Incorrect numerical values were used during the formula derivation, and 2) The existence of endpoints in the line segments was overlooked. By contrast, CoI easily identifies the number of intersection points in the image generated by SyMLLM.

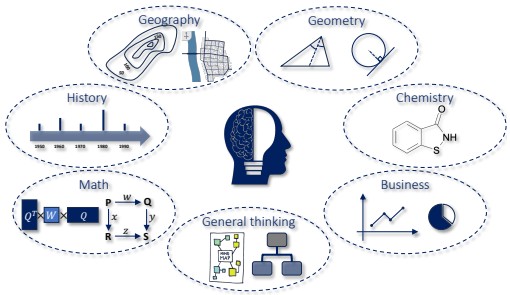 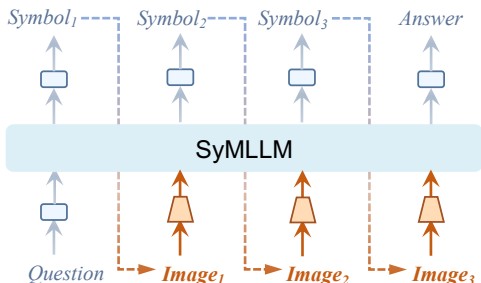

Figure 2: Images play a pivotal role in many disciplines. We tend to imagine pictures to solve problems intuitively.

Figure 3: The symbolic response generated by SyMLLM given language instruction can be directly converted to an image.

# 1 INTRODUCTION

Large-scale pre-trained language models (LLMs) (Vaswani et al., 2017; Devlin et al., 2018; Brown et al., 2020; Chowdhery et al., 2022; Touvron et al., 2023; OpenAI, 2023) have found applications in diverse areas. Beyond their robust foundational capabilities (Kaplan et al., 2020; Ouyang et al., 2022), a significant advancement lies in the evolution of prompting techniques (Liu et al., 2023b), which provide LLMs with a context to guide their responses and reasoning processes. As presented by Brown et al. (2020), the few-shot prompting approach offers the model several question-answer pairs to guide its outputs. The Chain-of-Thought (CoT) prompting strategy (Wei et al., 2022) provides sequential, step-by-step answer examples, enabling the model to perform intricate multi-step reasoning. The zero-shot CoT technique (Kojima et al., 2022) omits to provide examples, instead stimulating the model to think progressively with a simple "Let's think step by step" prompt. For instance, in Figure 1, to accurately count the intersection points of geometric shapes, CoT follows a sequential approach: 1) Determine the formula of each shape, 2) Compute the intersection points between each pair of shapes, and 3) Sum up all the intersection points.

However, the language-based reasoning process is often overly complex and abstract. Using images as a medium to solve textual logical problems is very intuitive. Humans excel at recalling visual concepts. For instance, one can easily recall someone's appearance while swiftly forgetting their name. The human brain is inherently adept at intuitive reasoning. We constantly employ imagery to facilitate thought processes in our daily lives. And many academic disciplines, as illustrated in Figure 2, utilize images to aid understanding and comprehension.

Drawing inspiration from the human brain's proficiency in intuitive reasoning, we have developed the Chain-of-Images (CoI) method to generate images to help solve complex and abstract problems. As illustrated in the right top of Figure 1, we generate images for the three shapes step by step. Starting from the second step, we identify the intersection points between the newly generated shape and the shapes from previous steps in the image. While algorithmically computing intersection points is challenging for LLMs, with the images, discerning the count of intersection points is very easy for an image encoder with good pattern recognition capabilities.

As a preliminary attempt, we point out that our Chain of Images method broadly offers at least but far more than these three advantages:

- Images provide an intuitive representation of the relationships (spatial, topological, temporal, etc.) between the items for tasks requiring complex relational reasoning, such as counting the intersection points of geometric shapes.

- Images assist in providing a compact representation that can record the current state at each step for tasks that satisfy Markovian properties and involve extended reasoning chains, like playing chess.

- Images capture a wealth of commonsense knowledge not present in language alone. Incorporating images into the reasoning process enhances its robustness.

```
Q: Is it more advantageous to address these tasks by generating images for a more intuitive solution? Why?
"In the following chess position, find a checkmate-in-one move. 1. d4 g6 2. c3 d5 3. Qd3 Bf5 4. Qe3 Nc6 5. h3 Nf6 6. g4 Be6
7. f3 Bg7 8. Nd2 Qd6 9. Kf2 h5 10. Qd3 hxg4 11. hxg4 Rxh1 12. Kg2 Rh4 13. Nh3 Bxg4 14. fxg4 Rxg4+ 15. Kh1 O-O-O 16. Nf3 Rh8."
A: Yes, it's often easier for many people to understand a chess problem by visualizing the position on a board rather than
parsing through a series of moves in descriptive notation. Generating an image of the board position can help a player
quickly identify threats, patterns, and potential moves.
Q: Is it more advantageous to address these tasks by generating images for a more intuitive solution? Why?
"There is a line segment from (-3.8, -1.1) to (1.7, -2.1). There is a polygon with coordinates [(-1.2, 3.6), (0.8, -3.2),
(4.6, 2.2)]. There is a circle centered at (-2.0, -0.2) with radius 3.0. How many intersection points are there?"
A: Yes, visual representation is definitely useful for problems like this, as it can quickly give a sense of the relative
positioning and potential intersections of the various geometrical shapes.
Q: Is it more advantageous to address these tasks by generating images for a more intuitive solution? Why?
"What is 4 plus 2?"
A: No, this problem is straightforward and can be easily solved without the aid of visual tools.
Q: Is it more advantageous to address these tasks by generating images for a more intuitive solution? Why?
"QUESTION"
A:
```

Figure 4: The 3-shot prompt for building the CoIEval dataset, from which the "QUESTION" should be replaced by the questions in each task.

In supporting our CoI reasoning, we introduce a symbolic multimodal large language model (SyM-LLM) shown in Figure 3 that generates images strictly based on language instructions and accepts both text and image as input. Furthermore, to validate the value of CoI, we have developed a CoI evaluation dataset, including 17 distinct tasks. This dataset is intended to assess how images can facilitate reasoning quantitatively.

Empirical results on geometric, chess, topological, temporal, and commonsense tasks from the CoI evaluation dataset demonstrate that the SyMLLM framework can generate the required images with nearly 100% accuracy. Based on such accurate image generation, SyMLLM employing CoI significantly outperforms the same-structure LLM that relies on pure text for reasoning on these tasks. For instance, the accuracy of calculating the number of intersection points among six shapes increased from 16.0% to 58.04%.

## 2    CHAIN OF IMAGES EVALUATION DATASET

Recent advances in Multimodal Large Language Models (MLLMs) such as those by Wu et al. (2023), Dong et al. (2023), and Zheng et al. (2023a) have significantly improved the generation of detailed images and comprehension of visual content. However, the integration of graphical imagination in reasoning processes remains a critical area of exploration. To assess and enhance MLLMs' ability to integrate text and image-based reasoning, we introduce the CoI Evaluation Dataset or CoIEval. This dataset explicitly measures MLLMs' proficiency in utilizing image generation as a part of textual reasoning processes. We utilize GPT-4 to sift through various evaluation datasets, including BIGBench (Srivastava et al., 2022), AGIEval (Zhong et al., 2023), OGB (Hu et al., 2020), MMLU (Hendrycks et al., 2020), and NYCC (Hessel et al., 2023), identifying questions amenable to CoI tasks. We created a specialized 3-shot prompt to guide GPT-4 in recognizing scenarios where visual aids can significantly enhance problem-solving. This prompt, illustrated in Figure 4, consists of three distinct questions highlighting the beneficial role of image generation in understanding complex spatial relationships, as demonstrated in chess and geometry, contrasted with more straightforward scenarios like arithmetic problems, where such aids are less impactful.

Applying the 3-shot prompt, we access a collection of 17 tasks across diverse domains, detailed in Appendix 6. We selected four datasets from this collection to empirically validate our CoI methodology. The chosen examples, representing different task types, are summarized in Table 1.

**Geometric**: The Geometric dataset comprises 250,000 examples  (Srivastava et al., 2022), each presenting various geometric shapes such as circles, line segments, and polygons, along with their coordinates. The task involves determining the number of intersection points based on the positions of these shapes. As depicted in Figure  7, the dataset categorizes problems into five levels of difficulty, ranging from 2 to 6 geometric shapes per question. The challenge of this task arises from the need for intricate calculations to determine the intersection points based on the positions of the shapes.

Table 1: Subset of the CoIEval dataset.

| | |
|---|---|
| Geometric | There is a line segment from (0.5, 3.4) to (3.5, -4.2). There is a polygon with coordinates [(0.5, 2.1), (-1.5, -3.0), (1.4, -6.3)]. There is a circle centered at (-0.7, -0.4) with radius 3.0. How many intersection points are there? |
| Chess | In the following chess position, find a checkmate-in-one move. 1. d4 g6 2. c3 d5 3. Qd3 Bf5 4. Qe3 Nc6 5. h3 Nf6 6. g4 Be6 7. f3 Bg7 8. Nd2 Qd6 9. Kf2 h5 10. Qd3 hxg4 11. hxg4 Rxh1 12. Kg2 Rh4 13. Nh3 Bxg4 14. fxg4 Rxg4+ 15. Kh1 O-O-O 16. Nf3 Rh8. |
| Topological | Could you please count the number of 6-cycles present in the following molecular graph? (0, 1), (1, 2), (2, 3), (3, 4), (3, 5), (5, 6), (6, 7), (7, 8), (8, 9), (9, 10), (10, 11), (10, 12), (3, 13), (13, 14), (14, 15), (12, 6). |
| Temporal | Question: Out of the following options, which time is closest to 03:21:15? Options: (A) 06:36:45, (B) 05:36:44, (C) 10:51:49, (D) 08:34:47. |

**Chess**: This task involves analyzing a series of chess moves provided in PGN (Portable Game Notation) format (Feng et al., 2023) to identify a checkmate in one move. "One move" denotes that each player has made a single move. A "checkmate" is a scenario where the king is in imminent danger of capture with no legal moves remaining for escape. The dataset is categorized based on the number of moves, divided into segments of 20 moves each. For instance, a game state with 33 moves would be categorized under the [21, 40) moves group. The complexity of this task arises from the intricate sequence of moves leading to the checkmate, demanding a comprehensive understanding of chess tactics and strategies.

**Topological**: This dataset is derived from the molecular structures of the OGBG-HIV virus (Hu et al., 2020), where each molecular structure is represented as a graph. In these graphs, each node signifies an atom, and a pair of atoms denotes a bond between the two atoms. The task involves identifying the number of 6-cycles within each molecular graph. A 6-cycle is a ring structure formed by six atoms connected end-to-end, and such structures are crucial in molecular chemistry, with benzene rings being a notable example. Identifying 6-cycles is a challenging task in the field of Graph Neural Networks (GNN) (Huang et al., 2022). Similarly, for Large Language Models (LLMs), detecting 6-cycles from a list of edges requires advanced topological reasoning abilities.

**Temporal**: This task involves presenting a specific moment in the 12-hour format (hh:mm:ss) and offering four alternative times as options. The objective is to select the option that is closest to the specified time. The challenge's complexity stems from the need for precise conversions between hours, minutes, and seconds, and calculating the differences between various times. It requires comparing which option is nearest to the given time while accounting for the cyclical nature of time. For instance, 00:01 AM is only a few minutes apart from 11:59, despite the difference in hours.

## 3 SYMBOLIC MULTIMODAL LARGE LANGUAGE MODELS

The foundational step in the CoI methodology involves step-by-step image generation. Existing MLLMs equipped with the ability to comprehend and create images, often using diffusion models as image decoders. However diffusion models face notable limitations in control and precision. To address these challenges, we present the Symbolic Multimodal Large Language Model (SyMLLM), illustrated in Figure 3. SyMLLM, akin to LLaVA (Liu et al., 2023a), consists of an LLM, an image encoder, and a MLP that connects them. Unlike LLaVA, which lacks image generation capabilities, SyMLLM can generate symbolic representations in response to textual prompts. These symbols are then rendered into bitmap images using CairoSVG[1], and the resulting images are transformed into embeddings by the image encoder. These embeddings are concatenated with text embeddings for subsequent token prediction. SyMLLM leverages the strength of LLMs in generating precise symbolic representations from language contexts and seamlessly converts these symbols into image formats. SyMLLM demonstrated nearly 100% accuracy in our experiments in generating abstract images, markedly outperforming diffusion model-based MLLMs. This advancement underscores the significance of SyMLLM's approach in combining precision and control in image generation within the MLLM framework.

---

[1]https://cairosvg.org/documentation/

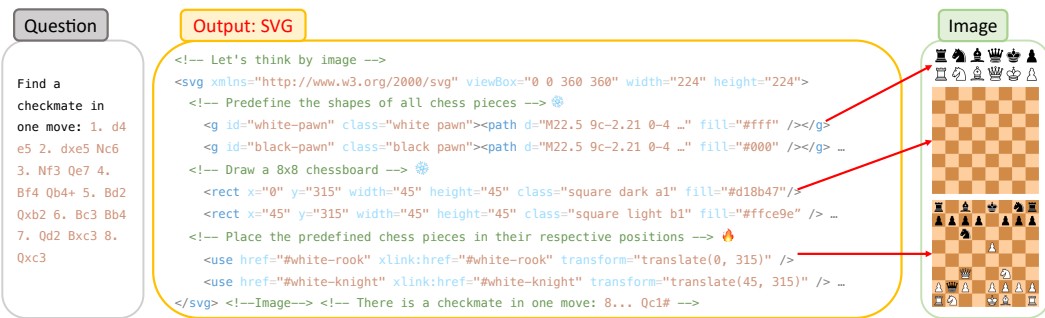

Figure 5: SVG format basic geometric shapes.

Figure 6: SVG format chess board and pieces.

The symbol we use in this paper is Scalable Vector Graphics (SVG). SVG is a widely adopted XML-based format for creating two-dimensional graphics. Unlike bitmap graphics which are made up of pixels, the raw file of SVG graphics are composed of language that can directly generated by LLMs. Meanwhile, they can be shown as pixel images scaled up or down without any loss in quality. SVG offers the capability to produce graphics that range from simple shapes and icons(e.g. geometric shapes, molecular graphs, or clocks) to highly intricate and detailed illustrations (e.g. chess board and pieces).

As shown in Figure 5, for the problem of counting the intersections of geometric shapes, the output of SyMLLM is a vector image in SVG format. The instructions are written as code comment in the vector image: $<!--Let's\ think\ by\ image-->, <!--Draw\ a-->, <!--Image-->, <!--which\ has\ 4\ intersection\ points\ with\ the\ previous\ shapes.-->, <!--Therefore,\ there\ are\ 0+4=4\ intersection\ points\ in\ total.-->$. When come to $<!--Image-->$ tokens, we convert the vector images into pixel images and then input them into SyMLLM's image encoder. Based on the images, it is easy to compute the intersection points of the geometric shapes.

As shown in Figure 6, for the problem of predicting checkmate in one move, SVG first defines classes for different chess pieces using polygons, then draws an 8×8 chessboard, and finally moves each piece to its final position. Since the first two steps are fixed for any chess position, we can provide them as prompts to the large model and only need to generate the final position of each piece. When $<!--Image-->$ tokens appear, we convert the current state of the chessboard into a pixel image, which is then input into SyMLLM's image encoder. Based on the current state, it is easy to predict whether a checkmate is possible in the next move.

## 4 EXPERIMENTS

In this section, we evaluate the effectiveness of the CoI approach using the four tasks outlined in Table 1 from our CoIEval dataset. We begin by fine-tuning our SyMLLM on the Geometry and Chess tasks to compare the maximum potential of text-based reasoning and CoI reasoning. It is important to note that CoI is an image-centered reasoning method and is not limited to the

Table 2: Results on the Intersect Geometric Dataset. Vicuna-7b* and SyMLLM-7b* were fine-tuned on the training set. SyMLLM-7b*† denotes using multiple images in the reasoning process. All other models employed a 3-shot in-context learning approach.

| Model | Img src | Img Acc | Number of shapes | | | | |
|---|---|---|---|---|---|---|---|
| | | | 2 | 3 | 4 | 5 | 6 |
| Mistral-7b | w/o Image | — | 20.74 | 14.32 | 11.25 | 6.99 | 5.19 |
| LLaMA-2-7b | w/o Image | — | 30.14 | 16.97 | 8.38 | 4.59 | 3.19 |
| Vicuna-7b | w/o Image | — | 21.77 | 15.76 | 10.78 | 6.78 | 5.38 |
| Vicuna-7b* | w/o Image | — | 90.75 | 48.25 | 27.75 | 23.25 | 16.0 |
| NExT-GPT | By SD1.5 | 0 | 8.1 | 4.2 | 0 | 0 | 0.6 |
| LLaVA-7b | Symbolic | 72.18 | 23.01 | 13.84 | 2.65 | 0.15 | 2.12 |
| SyMLLM-7b* | Symbolic | 100 | **96.25** | 81.5 | 54.75 | 43.75 | 25.5 |
| SyMLLM-7b*† | Symbolic | 100 | **96.25** | **89.25** | **78.75** | **74.27** | **58.04** |

use of SyMLLM. We employ GPT-4V for image generation and task completion across the four tasks, leveraging its in-context learning capabilities. This approach allows for rapid application of CoI to a wide range of tasks. In our supplementary materials A, we have also utilized Stable Diffusion XL(SDXL)(Podell et al., 2023) and DALL·E 3 for image generation and LLaVA/GPT-4V for image recognition. Our experimental results indicate that CoI when combined with diffusion-based models, holds potential in Commonsense reasoning tasks as well. This finding highlights the versatility of CoI in enhancing the reasoning capabilities of MLLMs across various tasks.

## 4.1 GEOMETRIC

**Data Processing**: For the Geometric task, we meticulously created a training set by selecting 10,000 examples from each level of difficulty within the Geometric Dataset. To ensure a robust evaluation, we also randomly sampled 400 examples from each difficulty level to form the evaluation set, taking care to prevent any overlap with the training set. The training set was formatted according to the specifications shown in Figure 5, whereas the evaluation set was preserved in its initial format.

**Model Structure**: Our experimental framework utilized Vicuna-7B-v1.5 (Zheng et al., 2023b) as the primary LLM, along with clip-vit-large-patch14 (Radford et al., 2021) serving as the image encoder. A linear layer was included to ensure proper alignment between the image embeddings and the LLM. In order to establish a baseline for our comparisons, we also trained another instance of Vicuna-7B-v1.5 with identical hyper-parameters and prompts but without incorporating images in either the training or inference stages. To facilitate a thorough evaluation and comparison, we incorporated additional models into our analysis, such as Mistral-7b (Jiang et al., 2023), LLaMA-2-7b (Touvron et al., 2023), NExT-GPT (Wu et al., 2023), and LLaVA (Liu et al., 2023a).

**Training Details**: Adhering to the methodology proposed by Dettmers et al. (2023), we implemented a 4-bit quantization on the backbone of the LLM and employed a rank-16 LoRA, primarily focusing on the training of the newly added parameters. This fine-tuning process involved not only the the projection layer and ViT, but also the LLM itself. We conducted the training on a single

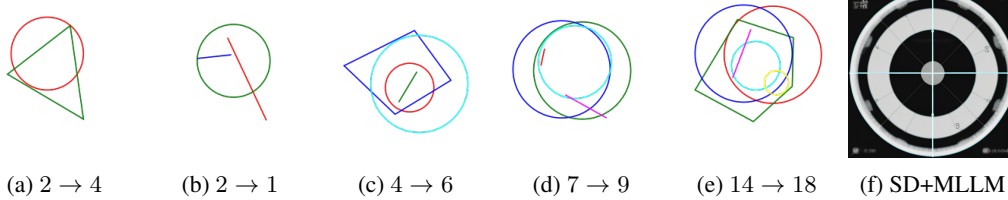

(a) 2 → 4     (b) 2 → 1     (c) 4 → 6     (d) 7 → 9     (e) 14 → 18     (f) SD+MLLM

Figure 7: Failure cases for 2 to 6 geometric shapes, respectively. The number preceding the arrow indicates the ground truth number of intersection points, while the number following the arrow shows the count by the SyMLLM.

Table 3: Results on Checkmate in One Move dataset.

| Model | Img src | Img Acc | Number of moves | | | | |
|---|---|---|---|---|---|---|---|
| | | | [1, 20) | [21,40) | [41,60) | [61,80) | [81,100) |
| LLaMA-2-7b | w/o Image | — | 9.39 | 6.39 | 5.56 | 3.27 | 4.55 |
| Vicuna-7b | w/o Image | — | 3.81 | 4.54 | 4.05 | 1.96 | 4.54 |
| RedPajama-3b | w/o Image | — | 1.27 | 5.15 | 5.05 | 1.31 | 0 |
| ChessGPT-3b* | w/o Image | — | 27.41 | 27.27 | 34.4 | 40.52 | 45.45 |
| NExT-GPT | By SD1.5 | 0 | 0 | 0 | 0 | 0 | 0 |
| LLaVA-7b | Symbolic | 57.9% | 0 | 0 | 0 | 0 | 0 |
| SyMLLM-3b* | Symbolic | 96.93 | 40.23 | 36.19 | 45.03 | 62.53 | 72.73 |
| SyMLLM-3b*† | Symbolic | 98.13 | **53.39** | **63.04** | **52.44** | **72.55** | **86.36** |

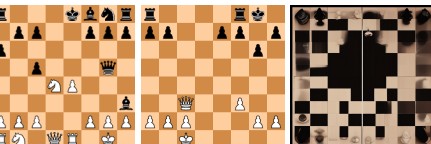 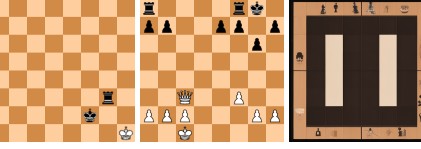

Figure 8: For two distinct move sequences, SyMLLM(Left) successfully generates the correct answers. In contrast, LLaVA(Center) consistently generates the SVG in the prompt, while NExT-GPT(Right) fail to produce accurate images.

A800 GPU, with a batch size set to 1 and using 8 gradient accumulation steps. The entire training spanned over 5 epochs and was completed in a total duration of 17.5 hours.

**Experiment Results**: From Table 2, The use of image generation before reasoning significantly surpassing the text-only method used by Vicuna-7b*. Our fine-tuned SyMLLM-7b* outperforms LLaMA, which utilizes a 3-shot symbolic prompt. Moreover, employing multiple images during inference further improves SyMLLM-7b*†, yielding better results than using a single image. Besides, NExT-GPT, using SD1.5, was unable to generate correct images thus the inference process is failed too. When provided with a 3-shot text to SVG prompt, LLaVA was capable of generating images with an accuracy of 72.18%. In contrast, our fine-tuned SyMLLM achieved nearly 100% accuracy in converting coordinates to images. The method for calculating image accuracy is based on the ratio of correctly generated geometric shapes to the total number of geometric shapes in each dataset entry. For instance, if a question involves four shapes and the model accurately generates three of them, the accuracy for that specific entry would be 75%. The overall image accuracy metric is then determined as the mean accuracy across all entries in the dataset.

**Results Analysis**: We have randomly selected one failure case from each level of difficulty and displayed it in Figure 7. It is evident that these failure cases are challenging to discern, even for the human eyes. One advantage of SVG lies in its scalability; it can be enlarged indefinitely without loss of quality. It is reasonable to believe that converting these graphics into larger images could further improve the performance of the CoI method on geometric intersection problems.

## 4.2 CHESS

**Data Processing**: Given the limited size of the dataset, it is solely utilized as a validation set. We derived our training dataset from a subset of data used in the research by Feng et al. (2023) on ChessGPT. This subset includes games from the Pro-player dataset[2] and the CCRL dataset[3]. We filtered this data to exclude games with non-standard openings, any illegal moves, or those not culminating in checkmate. The final numbers of training set are 4,015 (1-20 moves), 7,820 (21-40 moves), 6,659 (41-60 moves), 5,548 (61-80 moves), 2,886 (81-100 moves). Numbers of evaluation set are 788 (1-20 moves), 1,940 (21-40 moves), 593 (41-60 moves), 153 (61-80 moves), 22 (81-100 moves). The final moves, which represent checkmates, are labeled as answers, and the preceding moves as questions. This training set was then converted into SVG format as depicted in Figure 6.

---

[2]https://www.pgnmentor.com
[3]https://ccrl.chessdom.com

Table 4: In-context Learning experiments on Geometric, Chess, Topological and Temporal dataset.

| Model | Img src | Geometric | | Chess | | Topologic | | Temporal | |
|---|---|---|---|---|---|---|---|---|---|
| | | Img Acc | Acc | Img Acc | Acc | Img Acc | Acc | Img Acc | Acc |
| GPT-3.5 | — | — | 38% | — | 16% | — | 12% | — | 57% |
| GPT-4 | — | — | 39% | — | **66%** | — | 30% | — | 78% |
| GPT-4V | Symbolic | 99.3% | **49%** | 91.03% | 45% | 93.1% | **57%** | 97.3% | **87%** |

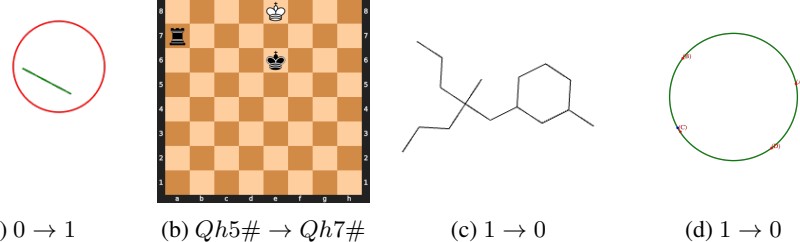

(a) $0 \rightarrow 1$    (b) $Qh5\# \rightarrow Qh7\#$    (c) $1 \rightarrow 0$    (d) $1 \rightarrow 0$

Figure 9: Failure case for In-Incontext learning

**Model Structure & Training Details**: The Backbone our SyMLLM used in this section is Chess-GPT (Feng et al., 2023), which is finetuned on the RedPajama-3B (Computer, 2023) using a comprehensive dataset of chess games. The training of this enhanced SyMLLM spanned over 5 epochs and was completed in a total duration of 11 hours. All other specifics are the same with the SyMLLM described in Section 4.1.

**Experiment Results**: As indicated in Table 3, the diffusion-based NExT-GPT failed to generate chess images, leading to ineffective reasoning. LLaVA, when provided with a 3-shot symbolic prompt, managed to generate legal chess boards, though with limited accuracy. In contrast, our SyM-LLM achieved an impressive accuracy of over 98%, significantly simplifying the task of predicting checkmate moves. The images generated by SyMLLM, LLaVA, and NExT-GPT are showcased for comparison in Figure8. Regarding question-answer accuracy, SyMLLM outperforms ChessGPT-3b, which was trained solely on text. The advantage of SyMLLM lies in its ability to predict the next move based on the final state's visual representation.

The performance of the SyMLLM on the "Checkmate in One Move" task is summarized , showcasing the CoI's high proficiency. The table indicates that SyMLLM achieves an accuracy rate near 100% in generating the correct chessboard state images. This accuracy facilitates the use of CoI reasoning to identify checkmate positions directly from the images, bypassing the need for complex textual reasoning chains. The CoI method significantly outperforms the baseline that relies on text-based inputs. For instance, the accuracy of [11,20) moves has improved to $1.38\times$, and the accuracy of [61,70) moves has increased to $1.85\times$.

**Results Analysis**: When the representation of the chessboard state is consistently accurate, the length of the reasoning chain becomes irrelevant. As the number of moves in a sequence increases, the pool of legal moves narrows, enhancing the likelihood of successfully identifying a checkmate in one move. This is corroborated by the results in the table, where an increase in the number of steps correlates with improved outcomes.

## 4.3 IN CONTEXT LEARNING

CoI can generate SVG images on GPT-4V using a 3-shot question-to-symbolic prompt approach. Subsequently, these images can be utilized for question answering through a 3-shot image-text answering method. In Table 4, we compare the performance of pure text-based GPT-3.5-turbo and GPT-4-1106-preview with GPT-4-1106-vision-preview, which generates images to assist in solving the four types of tasks mentioned in 6. It's important to note that due to OpenAI's limitations, GPT-4-1106-vision-preview can only process 100 requests per day, leading to each evaluation set containing only 100 data entries.

The data in the table indicates that, without any additional training, GPT-4V's textual capabilities are sufficient to generate accurate SVG images, though the improvement in results is not markedly significant. We visualized several failure cases in Figure9 and observed that even for straightforward tasks, GPT-4V's accuracy can be inconsistent. However, we believe that high-quality images generated through In Context Learning, combined with the next version of GPT-4V, have the potential to significantly enhance reasoning abilities across these four tasks.

## 5 RELATED WORK

### 5.1 REASONING WITH LLM

With the emergence of LLMs with surprisingly powerful capabilities, it has become possible to use language models to address various non-traditional natural language tasks, such as some reasoning tasks: solving arithmetic problems, deducting about the commonsense, playing a game (OpenAI, 2023; Srivastava et al., 2022; Qin et al., 2023), etc. This ability is first manifested in the series of GPT model (Radford et al., 2019; Brown et al., 2020), where the models address various tasks without fine-tuning by following the instruction of a paragraph of natural language called "*prompts*". Centered around this prompt-leading generation capability, a series of works have been proposed to enhance the reasoning ability of large models using different forms of prompts: Wei et al. (2022) introduced a chain-of-thought prompt which provides the model with learning examples of the target reasoning task containing the detailed reasoning steps, enabling the model to imitate the examples to generate answers to solve reasoning tasks—this capability is referred to as "*in-context learning*" capability. Further, simply based on tasks described in natural language by the model (rather than examples), it can also stimulate the model's reasoning ability—this is called a 'zero-shot prompt' (Kojima et al., 2022).

### 5.2 MULTIMODEL LARGE LANGUAGE MODEL

The underlying principle of this work involves integrating the pre-trained language model with another pre-train model for a specific modality, such as images. For instance, Li et al. (2023) proposes to train a lightweight Transformer to align the image representation from an image encoder with the textual representation from an LLM. Drawing inspiration from the instruction tuning techniques used in pure language models, Liu et al. (2023a) employs data generated by GPT-4 to enhance question-answering performance. Alayrac et al. (2022) achieves closer integration of image and text data by incorporating image information into the LLM through cross attention layer instead of embedding. Moreover, Girdhar et al. (2023) proposes a joint embedding across six different modalities, including images, text, audio, depth, terminal, and IMU data, without the need for training on the last four modalities. This is achieved by leveraging the zero-shot capacity of the large image-text models. The above-mentioned models do not possess image-generation capabilities. To make up for this shortcoming, Dong et al. (2023) realize image generation by combining the model proposed by Liu et al. (2023a) with the Stable diffusion (Rombach et al., 2022). Building on this, Wu et al. (2023) further integrates it with the model from Girdhar et al. (2023), utilizing its ability to align multiple modalities to generate and accept multiple modalities. Very recently, GPT-4 (OpenAI, 2023) has demonstrated its astounding ability to understand and generate in the modalities of images and voice. In the examples it showcased, the GPT-4 has already reached a level close to human capabilities, adding both confidence and pressure to the research in this field.

## 6 CONCLUSIONS

In this paper, we put forward a Chain-of-Image (CoI) prompting method, which imitates humans' ability to solve complex reasoning problems with the help of images. To accurately generate images to help the CoI reasoning, we propose the SyMLLM framework. We perform experiments on three kinds of datasets: Geometric, Chess and Commonsense, using the SyMLLM framework to demonstrate the power of CoI. We found that CoI uniformly improves the performance over pure-text models without images as an intermediate representation. We believe our CoI method has great potential to help LLMs achieve human-level reasoning ability.

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

Table 5: Results on the Commonsense Dataset.

|  | Location Accuracy(%) | Unusual Accuracy(%) |
|---|---|---|
| Text | 73.63 | 90.4 |
| CoI with SDXL | 77.78 | **100** |
| CoI with DALL·E 3 | **85.01** | **100** |

## A  COMMONSENSE

**Task Description**: Previous experiments have demonstrated that the Context of Inference (CoI) significantly aids in solving complex logical problems. In this section, we proceed to investigate whether images can enhance the capability of text in common sense reasoning. To this end, we have selected the Location and Unusual tasks for testing. These tasks utilized the same set of 531 event descriptions, but the questions differed. The Location task requires the description of an event, followed by a question about where the event took place. A model's answer is deemed correct only if it includes the provided standard answer. On the other hand, the Unusual task describes a scenario and then asks which parts of the description violate common sense. Given that questions about violations of common sense are open-ended, the accuracy of the answers cannot be directly calculated. Therefore, the study adopted different metrics: if the large-scale model fails to identify the nonsensical elements in the description, the answer is considered incorrect; otherwise, it is correct.

**Model Structure**: While the SyMLLM is adept at accurately rendering abstract graphics, the generative model creates richly detailed, realistic images. Therefore, we can use SDXL and DALL·E 3 to create common sense images without training a new model. We use LLaVA-13B, a model obtained by finetuning Vicuna-13B and ViT on a text-image dataset, to recognize these common sense images and answer questions. The pure text model we compare against is Vicuna-13B. The CoI method employs the prompt, "Let's solve the problem with the help of an image," to cue the model to use imagery for reasoning.

**Experiment Results**: As shown in the following table: the use of pictures as a medium can introduce visual common sense information to help improve common sense reasoning problems. For example, using SDXL to generate pictures can improve the accuracy of determining the location of an event by 4.15%. Using the more effective DALL·E 3 to generate pictures improves the accuracy by 9.66%. Using the plain text model, 9.6% of the samples could not find the anti-commonsense points in the textual descriptions, but it is easy to find them after using the pictures.

**Results Analysis**: As shown in Figure 10 and Figure 11, when solving the same problem, CoI is more intuitive compared to text-only reasoning and is able to supplement the missing details in the text with visual commonsense knowledge to aid the reasoning process. In addition, the strength of the image generation capability directly affects the reasoning efficiency of the CoI; for example, the CoI with DALL·E 3, which generates images of higher quality, is 4.51% more accurate in recognizing the location of an event than the CoI with SDXL.

## B  CoIEVAL

In this table, each row is divided into three lines: Line 1. The task name and the source dataset from which the task is derived. Line 2. An example question from the task. Line 3. GPT-4's response regarding whether image generation aids in reasoning—indicating 'Yes' or 'No', along with the rationale for why GPT-4 believes an image would or would not be helpful.

Checkmate in One @ BIGBench
Example: In the following chess position, find a checkmate-in-one move. 1. f4 Nc6 2. Nf3 Nf6 3. d4 d5 4. e3 Bd7 5. Be2 e6
GPT-4 Response: Yes, generating an image of the board position can help a player to quickly identify threats, patterns, and potential moves.

Chess State Tracking @ BIGBench
Example: For each of the following chess games, please complete the notation for the last shown move by filling in the destination square: d2d4 e7e6 e2e3 c7c5 f2f4 b7b6 g1f3 d7d6 f1e2 a7a6 e1g1 g7g6 c2c3 f8g7 c1d2 g8e7 d2e1 e8g8 e3e4 c8b7 b1d2 f7f5 d1b3 b7
GPT-4 Response: Yes, generating an image of the chess board would allow for a more intuitive understanding of the game's progression and the positions of the pieces.

Geometric Shapes @ BIGBench
Example: This SVG path element <path d="M 31.94,78.63 L 66.91,49.50 L 68.54,41.07 L 61.03,39.02 M 61.03,39.02 L 52.78,44.98 M 52.78,44.98 L 31.94,78.63"/>draws a [sector, heptagon, octagon, circle, rectangle, hexagon, triangle, line, kite, pentagon]
GPT-4 Response: Yes, an image would provide a clear, visual representation of the shape, making it easier to identify.

Intersect Geometry @ BIGBench
Example: Find the number of intersection points between the shapes and lines specified by the coordinates given. There is a line segment from (-0.1, -2.2) to (-2.4, -2.4). There is a circle centered at (1.8, 1.5) with radius 2.0. How many intersection points are there?
GPT-4 Response: Yes, by plotting the line and the circle on a graph, one can easily see where they intersect, especially for those who are more visually inclined.

Logic Grid Puzzle @ BIGBench
Example: There are 2 houses next to each other, numbered 1 on the left and 2 on the right. There is one person living in each house. The people in these houses have different characteristics: Each person plays a different sport: one is a soccer player and one is a baseball player. Each person has a different device: one has a radio and one has a gameboy. Clues: 1. The baseball player lives in the first house. 2. The person who has a gameboy does not live in the first house. What is the number of the house where the person who has a gameboy lives?
GPT-4 Response: Yes, it can help to visualize the positions of the houses and the characteristics of the people living in them, making it easier to solve the problem.

Logical Deduction @ BIGBench
Example: The following paragraphs each describe a set of five objects arranged in a fixed order. The statements are logically consistent within each paragraph. On a shelf, there are five books: a green book, a blue book, a white book, a purple book, and a yellow book. The blue book is to the right of the yellow book. The white book is to the left of the yellow book. The blue book is second from the right. The purple book is the second from the left.
GPT-4 Response: Yes, visualizing the arrangement of the books can make it easier to understand their relative positions and to answer questions about the order.

Matrix Shapes @ BIGBench
Example: Compute the kronecker product of a matrix of shape (2,2,3,3) with a matrix of shape (2,2,4,2). Add the result to a matrix of shape (4,4,12,6). Sum the result over the second axis. Compute the kronecker product of the result with a matrix of shape (2,3,3).
GPT-4 Response: Yes, generating images can help in visualizing the shapes and dimensions of the matrices involved, making it easier to solve the problem.

Movie Dialog Same or Different @ BIGBench
Example: The following is a conversation between two people, but the transcript doesn't mark who said what: "You're asking me out." "That's so cute." "What's your name again?"—— In the preceding conversation, were the sentences "That's so cute." and "What's your name again?" said by the same or different individuals?
GPT-4 Response: Yes, it can help to visually organize the conversation and make it easier to track who might be speaking at any given time.

Reasoning About Colored Objects @ BIGBench
Example: On the table, you see a bunch of objects arranged in a row: a grey cat toy, a brown paperclip, an orange envelope, a gold notebook, a red teddy bear, and a magenta mug. What is the color of the object directly to the right of the gold object?
GPT-4 Response: Yes, it allows for a more intuitive understanding of the spatial arrangement of the objects, making it easier to identify the object to the right of the gold object.

AQuA-RAT @ AGIEval
Example: A car is being driven, in a straight line and at a uniform speed, towards the base of a vertical tower. The top of the tower is observed from the car and, in the process, it takes 10 minutes for the angle of elevation to change from 45° to 60°. After how much more time will this car reach the base of the tower?
GPT-4 Response: Yes, visual representation can be helpful for this problem. Drawing a diagram with the tower, car's positions, and angles of elevation can make it easier to apply trigonometric concepts and visualize the distances and relationships between the car and tower.

LogiQA-EN @ AGIEval
Example: In the planning of a new district in a township, it was decided to build a special community in the southeast, northwest, centered on the citizen park.These four communities are designated as cultural area, leisure area, commercial area and administrative service area.It is known that the administrative service area is southwest of the cultural area, and the cultural area is southeast of the leisure area. Based on the above statement, which of the following can be derived?
GPT-4 Response: Yes, in cases like this where spatial relationships are described, a visual representation can be very helpful. By mapping out the relative positions of the communities, it can provide a clearer understanding of the arrangement and can assist in deriving accurate conclusions based on the provided information.

LSAT-AR @ AGIEval
Example: Of the eight students—George, Helen, Irving, Kyle, Lenore, Nina, Olivia, and Robert—in a seminar, exactly six will give individual oral reports during three consecutive days—Monday, Tuesday, and Wednesday. Exactly two reports will be given each day—one in the morning and one in the afternoon—according to the following conditions: Tuesday is the only day on which George can give a report. Neither Olivia nor Robert can give an afternoon report. If Nina gives a report, then on the next day Helen and Irving must both give reports, unless Nina's report is given on Wednesday.", "question": "Which one of the following could be the schedule of the students' reports?"
GPT-4 Response: Yes, for this kind of logic-based puzzle, visual representation like a chart or a table can be beneficial. It allows for easier organization of information and constraints, making the process of elimination and arrangement more intuitive.

High School Mathematics @ MMLU
Example: If a pentagon P with vertices at (− 2, − 4), (− 4, 1), (−1, 4), (2, 4), and (3, 0) is reflected across the line y = x to get a new pentagon, P', then one of the vertices of P' is
GPT-4 Response: Yes, by plotting the points and reflecting them across the line, it becomes easier to identify the new position of the vertices and determine which one corresponds to P'.

Topological @ OGBG-HIV
Example: Could you please count the number of 6-cycles present in the following molecular graph? (0, 1), (1, 2), (2, 3), (3, 4), (3, 5), (5, 6), (6, 7), (7, 8), (8, 9), (9, 10), (10, 11), (10, 12), (3, 13), (13, 14), (14, 15), (12, 6).
GPT-4 Response: Yes, manually counting 6-cycles based on the edge list alone would be error-prone and time-consuming. Creating a visual representation of the graph, such as a diagram or image, is a standard and practical approach for solving this type of problem.

Temporal
Example: Out of the following options, which time is closest to 03:21:15? Options: (A) 06:36:45, (B) 05:36:44, (C) 10:51:49, (D) 08:34:47.
GPT-4 Response: Yes, creating a clock image to address this task can indeed provide a more intuitive solution. Visualizing the clock hands can help us quickly determine which option is closest to 03:21:15 without performing complex calculations.

Location @ NYCC
Example: In a tree, a mother bird is tending to her nest and babies. There is another bird in a cage on an upper branch. Where can this event take place?
GPT-4 Response: Yes, the key of this problem is that the location should allow for trees where birds can naturally make nests and also accommodate a caged bird. An image could help illustrate this scenario, making it more tangible and perhaps easier to understand, especially for visual learners.

Unusual @ NYCC
Example: A group of people are crowded around in a living room talking to one another. A man in the foreground introduces two individuals one appears to be a regular human male the other appears to be an animal. What is unusual about this description?
GPT-4 Response: Yes, an image can add context and may help highlight the oddity of an animal being introduced as if it were a person, which is the unusual aspect of this description.

Table 6: Datasets that GPT-4 believes generate images would be helpful.

```
Question: A woman is waiting to get on the elevator. But the people in the elevator
are on fire. Where can this event take place?
Answer: An elevator
Text Only: This event can take place in a dream or a surrealistic
setting, as it is a highly unusual and unlikely scenario.
CoI(SDXL): Let's solve the problem with the help of an image. This
event can take place in a building, specifically in an elevator.
The image shows a woman standing in front of an elevator with flames
coming out of the elevator's doors. This situation could occur due
to a malfunction or an accident, such as an electrical short-circuit
or a fire caused by an external source.
CoI(DALL·E 3): Let's solve the problem with the help of an image.
This event can take place in a building with an elevator, as seen in
the image. The elevator has a fire-themed design, and the doors are
open, revealing a group of people inside. The woman standing outside
the elevator is looking at the people inside, who appear to be
dressed in business attire. This suggests that the event might be
taking place in a professional or corporate setting, such as an
office building or a conference center.
```

Figure 10: This is an example for the same question, the response of Text only reasoner, Chain of images with original picture and CoI with image generated by stable diffusion. CoI improves the performance of this problem

```
Question: A group of people are crowded around in a living room talking to one another.
A man in the foreground introduces two individuals one appears to be a regular human
male the other appears to be an animal. What is unusual about this description?
Answer: There is a hairy, furry animal standing upright in a living
room with other human beings inside.
Text Only: There is nothing unusual about a man introducing an animal
to another individual in a living room. This could imply that the
situation is not extraordinary or out of the ordinary.
CoI(SDXL): Let's solve the problem with the help of an image. The
unusual aspect of the description is that it describes a man
introducing two individuals, one of whom is an animal. This is not a
typical scenario in everyday life, as animals are not typically present
in social gatherings or introductions. However, in the image, there is
a group of people crowded around in a living room, and one man is
introducing two individuals, one of whom appears to be an animal.
CoI(DALL·E 3): Let's solve the problem with the help of an image. The
unusual aspect of the image is the presence of an anthropomorphic fox
character standing and interacting with human characters as if it were
a normal occurrence. The fox stands on two legs, is dressed in human clo
gestures with its hand in a human-like manner.
```

Figure 11: This is an example for the same question, the response of Text only reasoner, Chain of images with original picture and CoI with image generated by stable diffusion. CoI improve the performance of this problem

