# OpenReview forum: "Chain of Images for Intuitively Reasoning"
_ICLR.cc/2024/Conference — Submitted to ICLR 2024_

### Official Review · Reviewer_ZLmV · 2023-10-15

**Soundness:** 3 good
**Presentation:** 3 good
**Contribution:** 2 fair
**Rating:** 5
**Confidence:** 3

**Summary:**

The paper presents Chain of Images (CoI), which generates images as intermediate representations and inserts them into complex language reasoning problems.
An image can represent complex textual logic in a more compact and intuitive way.
Thus, the newly added visual intuition eliminates the textual hallucination problem and introduces visual commonsense knowledge, thus enhancing logical reasoning for current large language models.

**Strengths:**

1. The paper proposes to build a symbolic multi-modal model (SyMLM) that can strictly generate images following language commands.
The proposed method assists large language models in reasoning.
2. Experiments on Geometry, Chess, and Common Sense tasks show the effectiveness of Chain of Images prompting compared with pure-language Chain of Thoughts (CoT) baselines.
3. The paper is well-written and easy to read.

**Weaknesses:**

1. The novelty is relatively limited.
There are some similar methodologies for multi-modal large language models to generate images(related work Sec.4.2). The paper is more concerned about using multi-modality to assist language reasoning. The claimed novelty is not that strong -- it is a less-studied task, rather than a new methodology.

2. The method is difficult to extend to broader applications.
    - The experiments are mainly done on Geometry and Chess task, which is very simplified tasks.
    - The setting of commonsense tasks is confusing. It seems that the image is generated by stable diffusion instead of the proposed SyMLM. Thus, it seems not a good evaluation for the proposed method.
    - To generate the Chain of Images, extra training is needed for each task. This largely hurts the universal capability of large language models and hinders broader application. Also, given the diverse reasoning tasks large language models can solve, the experiments cannot show the effectiveness of the method on other reasoning tasks.

**Questions:**

Questions have been mentioned in "Weakness" section.

---

> ### Author Response · Authors · 2023-11-22
> **To Reviewer ZLmV,**
>
> Thank you for your valuable feedback. We have revised our paper accordingly and will address each of your concerns:
>
> 1. **Novelty of the Methodology:**
>     - Unlike MLLMs that focus primarily on solving VQA tasks with a given ground truth image, our Chain of Images (CoI) method tackles more complex tasks by firstly generating images from textual queries. This approach goes beyond merely aligning image features with textual contexts, as is common in VQA tasks. CoI directly incorporates images into the reasoning process, representing a more advanced multimodal reasoning form, thereby enhancing the model's capabilities beyond text-only analysis.
>     - The SyMLLM framework is different from other MLLMs like NExT-GPT, DreamLLM, and MiniGPT-5, which use diffusion models for pixel image generation. SyMLLM excels in creating vector images, offering enhanced control and suitability for complex reasoning tasks, thereby providing a novel approach for multimodal generation models.
> 2. **Complexity and Variety of Experiments:**
>     - In addition to the Geometric dataset, we have conducted experiments in Chess, Commonsense, Topological, and Temporal tasks. These tasks, typically challenging for text-only large models, are significantly simplified when converted to images, demonstrating the valuable problem-solving capabilities of the CoI method.
> 3. **Commonsense Task Setting:**
>     - CoI primarily focuses on image-centered reasoning and is not solely dependent on SyMLLM. We have conducted experiments with GPT-4V and LLaVA across four tasks, and explored the combination of Stable Diffusion XL and DALL·E 3 for image generation with LLaVA/GPT-4V for image recognition. This approach allows us to evaluate CoI's effectiveness in commonsense datasets.
> 4. **Training Requirements and Universal Capability:**
>     - Our experiments, as shown in Table 3, demonstrate that in-context learning can effectively generate complex images without extensive training. This proves that our method can maintain the universal capability of large language models and does not require significant additional training for each task.
>
> We look forward to your response and are willing to answer any further questions you may have.

---

### Official Review · Reviewer_bdXh · 2023-10-22

**Soundness:** 2 fair
**Presentation:** 2 fair
**Contribution:** 2 fair
**Rating:** 3
**Confidence:** 4

**Summary:**

The paper presents Chain of Images (CoI), a method that simplifies complex language reasoning tasks into pattern recognition problems by generating a sequence of intermediate images.

Additionally, it introduces the Symbolic Multi-Modal Model (SyMLM), which strictly generates these images based on language instructions.

Experiments on three real-world datasets demonstrate that CoI outperforms language-only Chain of Thoughts (CoT) baselines.

**Strengths:**

I concur with the central thesis of the paper, which advocates for the utilization of visual rationales to enhance existing language-only chain-of-thought methodologies. The exposition of the intuition is lucid, and Figure 1 effectively illustrates the concept.

The paper excels in devising a method to map textual inputs to visual rationales and subsequently using these visual intermediates to facilitate reasoning. This approach constitutes the paper's primary focus and is a significant contribution to the field.

**Weaknesses:**

1. The submission guidelines specify a maximum of 9 pages for the main text, yet this paper comprises slightly over 8 pages. This brevity raises concerns about the completeness and readiness of the work for publication.
2. In Section 2.3 ("Converting Symbols To Image"), the exposition on SVG and FEN formats is overly succinct. It is unclear to the readers how these formats facilitate the effective translation of symbols into images.
3. Regarding Figure 5:
- The manuscript does not elucidate how the question text "a line segment from (0.5, 3.4) to (3.5, -4.2)" translates into SVG format. Is there a specific prompt used for this translation?
- More importantly, the figure directly presents intersection points as intermediate answers without explaining the rationale or method behind their determination.
4. Table 1 suffers from a lack of detailed analysis:
- The term "geometric shapes in each sample, representing the different difficulty levels" is ambiguous. Providing illustrative examples would be beneficial.
- While the comparison between "CoI Acc" with and without "Img" suggests the effectiveness of "CoI Acc," it's puzzling why "Img Acc" is nearly 100%, yet "CoI Acc" remains low.
- Importantly, there is a lack of clarity on the methodology used for calculating the similarity between the image generated by SyMLM and the ground truth.
5. Pertinently, the experiments on the CommonSense dataset appear to test the compositional reasoning ability of LLaVA-13B rather than the efficacy of your proposed method, SyMLM. This creates a disconnect in the paper's narrative.

**Questions:**

1. In Section 2.2, the paper states, "Subsequently, these images are converted into image embeddings by the image encoder. The embeddings are then concatenated with the text embeddings to generate the next token." I wonder if the image embeddings can be directly concatenated with the text embeddings without employing any adapter layers or fine-tuning mechanisms.
2. Also in Section 2.2, you claim that "In subsequent experiments, we observe that the accuracy of image generation approaches nearly 100%." Could you please elaborate on the metrics used to quantify the similarity between the generated and target images?
3. In Section 2.3, the text mentions, "The image is then used to count intersection points." Could you specify the methodology employed for counting the intersection points within an image in the context of this work?

---

> ### Author Response · Authors · 2023-11-22
> **To Reviewer bdXh,**
>
> Thank you for your detailed review. Here are the responses to your concerns:
>
> 1. **Length of the Paper:**
>     - We have rewritten the paper to enhance the reliability of our experiments and improve readability. This revision ensures comprehensive coverage of our research within the specified page limit.
> 2. **Explanation on SVG and FEN Formats:**
>     - In all datasets, we uniformly utilize the SVG format to boost the method's universality. SVG files can be directly displayed on screens or easily converted into formats like PNG or JPEG using tools like CairoSVG.
> 3. **Clarification on Figure 5 and Intersection point recognization:**
>     - The SVG format follows a fixed syntax, which can be generated using prompts. The coordinates for SVG elements like cycles, lines, and polygons are readily extracted from the question text.
>     - Once a question is converted into an image, it becomes easy for both human observers and our SyMLLM to identify elements like intersection points at a glance.
> 4. **Detailed Analysis in Table 1:**
>     - Thank you for your suggestion. In the revised paper, we have thoroughly addressed the three issues you raised, providing a more detailed analysis in the revised paper.
> 5. **Experiments on the CommonSense Dataset:**
>     - Our CoI method is not restricted to using SyMLLM exclusively; any model capable of generating images and utilizing them for reasoning is suitable. While SyMLLM excels at solving complex, abstract reasoning tasks, we also explored its effectiveness in general commonsense questions. Based on your advice, we moved this experiment to the supplementary material, focusing the main text on experiments using SyMLLM, thus maintaining consistency in the narrative.
> 6. **Image and Text Embedding Concatenation:**
>     - SyMLLM's architecture, similar to LLaVA, includes an image encoder, an LLM, and a projection layer. SyMLLM is distinct in its ability to generate precise images from textual instructions, thanks to fine-tuning on text-to-symbolic datasets.
> 7. **Metrics for Image Generation Accuracy:**
>     - In the revised version of the paper, we have redefined the metrics used. Specifically, for the geometric dataset, accuracy is calculated as the number of correctly predicted shapes per data point divided by the total number of shapes. For the chess dataset, accuracy is determined by the number of correctly predicted chess pieces divided by the total number of positions.
> 8. **Methodology for Counting Intersection Points:**
>     - For image encoding, identifying intersection points in an image requires basic pattern recognition capabilities. An intersection point is identified by the crossing of two different colors. This method is efficiently integrated into our model's processing capabilities.
>
> We look forward to your response and are willing to answer any further questions you may have.

---

### Official Review · Reviewer_QAod · 2023-10-31

**Soundness:** 3 good
**Presentation:** 2 fair
**Contribution:** 2 fair
**Rating:** 5
**Confidence:** 4

**Summary:**

This paper suggests Chain of Images (CoI), an intuitive method to improve Large Language Model(LLM) and Vision Language Model(VLM)’s complex reasoning ability. In contrast to Chain of Thoughs (CoT) which generates intermediate language descriptions for solving reasoning problems, CoI generates a series of images which serves as an intermediate representation, enhancing the model’s reasoning capabilities in domains where visual interpretation can be helpful. CoI is implemented as a symbolic multi-modal model (SyMLM), which directly generates symbolic representations of images from language instructions and uses both image and text as input. The authors tested their method in three different domains : geometry, chess, and common sense, and have shown that the integration of images to texts achieves better reasoning capabilities.

**Strengths:**

1. The idea is well motivated and simple, as humans intuitively imagine and reason using images in a range of domains.
2. The proposed method showed strong results compared to baselines in the tested domains.

**Weaknesses:**

1. Limited experiments
     - The tested domains are domains in which image generation should intuitively help. However, it is plausible that there are domains where incorporating image is not significantly helpful (e.g. arithmetics). It would be better if the authors have tried their method in more diverse domains.
     - In domains other than geometry, the LLM is generating only a single image. The authors should try more domains in which chain of image generation is needed. Moreover, in the geometric domain, the authors could compare their method against a simple baseline which generates the final image at once without chaining process.
     - As the proposed method is proposing to also use image along with text, the authors should have tried more extensive experiments using VLM using / not using their method.

2. Method is limited
     - Using symbolic representation (SVG, FEN) for image generation is domain specific. Moreover, assuming such representation to be given is a strong assumption to make.
     - Rather than fine-tuning LLM to generate images given a set of problems and corresponding images to be given as a training data, the authors can consider reducing the size of training data and simply try in-context learning.
   - Most of the results are either based on the perfect image generation ability of LLM after fine-tuned, or ground truth image.


3. Clarity of writing
     - Although the authors included examples of common sense domain, it is not easy to understand the two tasks tested in common sense reasoning. For instance, the first task of determining the scene described in the text can be misinterpreted as a task in which VLM determines the best image given a set of single text description and multiple images.
     - The authors didn't clearly state the results of a second task in common sense domain. The paper only contains two examples in Fig. 6 and Fig. 7.

**Questions:**

1. Can you elaborate the results of the second task in common sense domain? Current version of the paper only states "while CoI can identify something unusual in all images".

---

> ### Author Response · Authors · 2023-11-22
> **To Reviewer QAod,**
>
> Thank you for your insightful comments. We have addressed your points as follows:
>
> 1. **Applicability in Various Domains:**
>     - Regarding arithmetic, it is true that LLMs can potentially solve these problems in text space. Our experiments, however, focus on domains like Geometric, Chess, Topological, and Temporal, demonstrating the potential for more diverse applications of our method in more complex tasks.
> 2. **Need for Chain of Image Generation in Additional Domains:**
>     - We have compared the use of single v.s. multiple images in Tables 1 and 2. The results clearly indicate that using multiple images, especially in increasingly complex tasks, offers significant benefits over a single image.
> 3. **Experiments with Visual Language Models (VLMs):**
>     - VLMs typically solve VQA tasks which necessitate a ground truth image for answering questions about the image. Our CoI method, in contrast, is designed for more complex tasks where no ground truth image exists. We also explored the integration of VLMs with our text-to-symbolic prompts, converting them to a SyMLLM without finetuning. This is detailed in Tables 1 and 2, where LLaVA was able to generate relevant images, but struggled with abstract imagery.
> 4. **Use of Symbolic Representation (SVG, FEN):**
>     - SVG is not a domain-specific format but a widely-used XML-based format for creating vector images. In the new revised version paper, our experiments across various domains all utilize the SVG format, demonstrating its versatility.
> 5. **In-context Learning:**
>     - We have conducted in-context learning experiments, as shown in Table 3, highlighting the potential effectiveness of our method with reduced training data. And observing that GPT-4V can generate images using 3-shot text-to-symbolic prompts, although they struggle with visual questions, leading to unsatisfactory outcomes.
> 6. **Results Based on Finetuned Image Generation Ability:**
>     - In Table 3, we also demonstrate the accuracy of images generated through in-context learning, showcasing that satisfactory results can be achieved without relying solely on perfect image generation or ground truth images.
> 7. **Clarification on Common Sense Dataset:**
>     - We have revised the section on the common sense dataset for clarity. The Location task is evaluated based on multiple-choice answers, where the inclusion of standard answers determines correctness. The Unusual task involves open-ended questions about common sense violations. Here, we adopted different metrics: if a large-scale model fails to recognize nonsensical elements in the description, the answer is deemed incorrect; otherwise, it is considered correct.
>
> We look forward to your response and are willing to answer any further questions you may have.

---

> ### Comment · Reviewer_QAod · 2023-11-23
> **Response to Authors**
>
> Thanks for your comments. I have read the response and most of my concerns are addressed, and I will increase my score.
>
> In the meanwhile, I still question the scalability of SVG format, as it is mostly used for generating simplified images. The authors have included stable diffusion for realistic image generation, but such generative models were only helpful for common sense tasks, not in chess / geometry. The selection of appropriate generation of images still remains as a critical point to deal with.
>
> Additionally, I suggest the authors to improve the clarity of writing. For instance, it would be better to clarify the difference between using multiple images and single image for CoI for the readers without context.

---

### Official Review · Reviewer_KgMJ · 2023-11-05

**Soundness:** 2 fair
**Presentation:** 2 fair
**Contribution:** 3 good
**Rating:** 5
**Confidence:** 4

**Summary:**

The paper proposes a novel method named Chain-of-Image (CoI) prompting, which aims to enhance the reasoning abilities of large language models (LLMs) by incorporating a chain of generated images as intermediate representations. The method leverages a symbolic multi-modal model (SyMLM) that can transform textual prompts into SVG format images, which are then used to support the text-based reasoning process. The authors conduct experiments across various tasks, including geometry, chess, and commonsense reasoning, demonstrating that the CoI method outperforms text-only LLM baselines. The concept is innovative; however, the experiments may need further refinement to more robustly substantiate the claims. The scores maybe raised if the main concerns are addressed.

**Strengths:**

1. The approach of integrating a visual reasoning chain into the operation of LLMs is quite innovative. It mimics human problem-solving strategies, which often involve visualizing concepts and steps. From the figures, it's very intuitive that such a system may bring the reasoning process closer to human when perform complex reasoning tasks.
2. The paper appears to be methodologically sound, with a clear and well-organized description of the experimental setup, datasets, and baseline comparisons. Details to reproduce the paper are also clearly illustrated.
3. If the proposed method is as effective as the paper suggests, it has the potential to significantly impact how LLMs are used for complex problem-solving tasks, making them more accurate and versatile. Particularly in the domain of reasoning and natural language understanding, it can make LLMs more interpretable and powerful in tasks that require complex reasoning.

**Weaknesses:**

1. Lack of experiments comparing the diffusion-based model generated images vs the proposed more controllable image generation strategy. For example, a qualitative comparison can be shown in Figure 4 to show how the proposed method solve the issues presented. Besides, in Table 1/2, the ablations should also be shown to support the claim.
2. In contribution, it says "counting the intersection points of geometric shapes, images provide an intuitive representation of the relationships (such as spatial, topological, temporal, etc.) between the items", but seems like only the spatial relationship is the primary point that's been validated in the the experiment 3.1
3. There also lack important experiments, seems that only text-based baseline are included, how about compare with other multimodal large language models? is the CoI method still important?
4. The details of the whole multimodal language model is lacked. Especially in Section 2.2, there supposed to be some vision encoder details and the training details. Although in Section3.2, it mentions clip-vit-large-patch14, the whole model details are missing.

**Questions:**

1. How is the training/testing data generated, it's quite confusing. E.g., "We convert all of the 50,000 training samples to the form introduced in Section 2.3, while keeping the test set unchanged." What are the details of this conversion. And how about the n_train and n_test in Table 2. What are the statistics details in Table 1.
2. Is the whole pipeline a LLaVA-based model? what is the detail of the text-image connector.
3. How does this method compare with LLaVA? Is this method also benefiting multimodal large language models? or only get some advantage when compared with LLMs? Then is that really a fair comparison?

---

> ### Author Response · Authors · 2023-11-22
> **To Reviewer KgMJ,**
>
> Thank you for your constructive feedback. We have revised our paper according to your advice and will address your concerns point by point:
>
> 1. **Experiment Comparison Between Diffusion-Based Models and Our Controllable Image Generation Strategy:**
>     - We have compared the performance of NExT-GPT, a diffusion-based model, in Geometric and Chess tasks, as illustrated in Tables 1 and 2. The generated images and their respective analyses are detailed in Sections 4.1 and 4.2 of the revised paper. We assert that diffusion-based models are less effective in aiding abstract problem-solving.
> 2. **Validation of Different Types of Relationships in Experiments:**
>     - We have conducted experiments in Geometric, Chess, Topological, and Temporal tasks. The results, presented in Table 3, demonstrate CoI's efficacy in these areas, substantiating its capability to intuitively represent various relationships (spatial, topological, temporal, etc.) in problem-solving contexts. In the main paper, we have analyzed several failure cases and observed that GPT-4V demonstrates accurate capabilities in generating SVG images. However, the visual capabilities of GPT-4V are not as satisfactory, even in some simple examples. We believe that with the enhancement of visual processing abilities in large-scale models, there will be a significant improvement in this comparative outcome.
> 3. **Comparison with LLaVA-Based Models and Other Multimodal LLMs:**
>     - SyMLLM's architecture shares similarities with LLaVA, including an image encoder, an LLM, and a projection layer. However, unlike LLaVA models which are limited to image interpretation, SyMLLM is fine-tuned on text-to-symbolic datasets, enabling precise image generation from textual instructions.
>     - For LLaVA-like MLLMs, their primary function is solving VQA tasks requiring a ground truth image. In contrast, CoI tackles more complex tasks by firstly generating images from textual queries. We observed that LLaVA-like MLLMs, when combined with our text-to-symbolic prompts, could adapt to SyMLLM without fine-tuning. This finding, detailed in Tables 1 and 2, shows that LLaVA can generate relevant images, surpassing diffusion-based models in this aspect.
>     - We have also compared our method with NExT-GPT, capable of image generation using stable diffusion 1.5. The results, presented in Tables 1 and 2, indicate it's failed in both image generation and task-solving capabilities compared to CoI.
> 4. **Details of the Multimodal Language Model and Data Generation:**
>     - The detailed methodologies for Data Processing, Model Structure, and Training for each task are described in Section 4 of the revised paper.
>
> We look forward to your response and are willing to answer any further questions you may have.

---

### Author Response · Authors · 2023-11-22
**Dear All Reviewers,**

We appreciate your insightful feedback and have thoroughly revised our paper, incorporating your valuable suggestions. Here, we would like to highlight the key contributions of our revised paper:

1. Our work introduced the Chain of Images (CoI) method, distinct from VQA tasks that traditionally respond to queries about given images. CoI is designed for more complex tasks: for text-based reasoning problems where no predefined images are available. It innovatively generates images in response to a question, facilitating problem-solving through an integrated text-image approach. Additionally, traditional MLLMs employed in VQA tasks typically focus on aligning image features with textual spaces. This approach still leverages the text reasoning capabilities of LLMs. In contrast, our CoI method innovatively incorporates images directly into the reasoning chain. This represents a more advanced form of multimodal reasoning, thereby elevating the model's capabilities beyond conventional text-only analysis.
2. A significant challenge in CoI is the generation of precise images. To overcome this, we introduced the SyMLLM framework, a unified model for image generation and understanding. SyMLLM's architecture, inspired by LLaVA (or GPT-4V), incorporates an image encoder, a LLM, and a projection layer connecting them. Unlike LLaVA-like models that can only interpret images, SyMLLM is adept at generating images accurately, thanks to its fine-tuning on text-to-symbolic datasets. This approach is in contrast to other MLLMs like NExT-GPT, DreamLLM, and MiniGPT-5, which rely on diffusion models for pixel image generation. SyMLLM demonstrates superior performance in creating vector images, offering greater control and suitability for intricate reasoning tasks. We utilize SVG format in all the four experiments for its simplicity and effectiveness in vector image representation.
3. We initially fine-tuned SyMLLM on Geometry and Chess tasks, with the results presented in Tables 1 and 2. The use of generated images as an intermediary in problem-solving yielded a 3.6x improvement over text-only methods. Importantly, CoI is an image-centered reasoning method, thus it is not exclusively reliant on SyMLLM. We conducted experiments with GPT-4V and LLaVA across four tasks (Table 3), observing that these models can generate images using 3-shot text-to-symbolic prompts, although they struggle with visual questions, leading to unsatisfactory outcomes. Additionally, we explored combining Stable Diffusion XL and DALL·E 3 for image generation with LLaVA/GPT-4V for image recognition, to assess CoI's efficacy in commonsense datasets. Further details on these experiments are provided in Section 4 and Appendix of the revised paper.

*Table 1: Results on the Intersect Geometric Dataset. Vicuna-7b* and SyMLLM-7b* were fine-tuned
on the training set. SyMLLM-7b*† denotes using multiple images in the reasoning process. All other
models employed a 3-shot in-context learning approach.*
| Model | Image source | Image Accuracy | 2 shapes | 3 shapes | 4 shapes | 5 shapes | 6 shapes |
| --- | --- | --- | --- | --- | --- | --- | --- |
| Mistral-7b | w/o Image | — | 20.74 | 14.32 | 11.25 | 6.99 | 5.19 |
| LLaMA-2-7b | w/o Image | — | 30.14 | 16.97 | 8.38 | 4.59 | 3.19 |
| Vicuna-7b | w/o Image | — | 21.77 | 15.76 | 10.78 | 6.78 | 5.38 |
| Vicuna-7b* | w/o Image | — | 90.75 | 48.25 | 27.75 | 23.25 | 16.0 |
| NExT-GPT | By SD1.5 | 0 | 8.1 | 4.2 | 0 | 0 | 0.6 |
| LLaVA-7b | Symbolic | 72.18 | 23.01 | 13.84 | 2.65 | 0.15 | 2.12 |
| SyMLLM-7b* | Symbolic | 100 | 96.25 | 81.5 | 54.75 | 43.75 | 25.5 |
| SyMLLM-7b*†  | Symbolic | 100 | 96.25 | 89.25 | 78.75 | 74.27 | 58.04 |


*Table 2: Results on Checkmate in One Move dataset.*
| Model | Image source | Image Accuracy | [1,20) moves | [21, 40) moves | [41, 60) moves | [61, 80) moves | [81, 100) moves |
| --- | --- | --- | --- | --- | --- | --- | --- |
| LLaMA-2-7b | w/o Image | — | 9.39 | 6.39 | 5.56 | 3.27 | 4.55 |
| Vicuna-7b | w/o Image | — | 3.81 | 4.54 | 4.05 | 1.96 | 4.54 |
| RedPajama-3b | w/o Image | — | 1.27 | 5.15 | 5.05 | 1.31 | 0 |
| ChessGPT-3b* | w/o Image | — | 27.41 | 27.27 | 34.4 | 40.52 | 45.45 |
| NExT-GPT | By SD1.5 | 0 | 0 | 0 | 0 | 0 | 0 |
| LLaVA-7b | Symbolic | 57.9 | 0 | 0 | 0 | 0 | 0 |
| SyMLLM-3b* | Symbolic | 96.93 | 40.23 | 36.19 | 45.03 | 62.53 | 72.73 |
| SyMLLM-3b*†  | Symbolic | 98.13 | 53.39 | 63.04 | 52.44 | 72.55 | 86.36 |

*Table 2: In-context Learning experiments on Geometric, Chess, Topological and Temporal dataset.*
| model | Image source | Geometric Img Acc | Geometric Acc | Chess Img Acc | Chess Acc | Topological Img Acc | Topological Acc | Temporal Img Acc | Temporal Acc |
| --- | --- | --- | --- | --- | --- | --- | --- | --- | --- |
| gpt-3.5-turbo | — | — | 38% | — | 16% | — | 12% | — | 57% |
| gpt-4-1106-preview | — | — | 39% | — | 66% | — | 30% | — | 78% |
| gpt-4-1106-vision-preview | Symbolic | 99.3% | 49% | 91.03% | 45% | 93.1% | 57% | 97.3% | 87% |

---

### Meta-Review · Area_Chair_YZU5 · 2023-12-14

**Metareview:**

This submission introduces the Chain of Images (CoI) method, an innovative approach to enhance the reasoning abilities of large language models (LLMs) by incorporating a series of generated images as intermediate representations. The authors propose the SyMLLM framework, distinct from typical Visual Question Answering (VQA) tasks, to facilitate complex text-based reasoning through an integrated text-image process. This concept, alongside the comprehensive utilization of SVG format across experiments, demonstrates the novelty and potential impact of the work in the field of multimodal reasoning.

However, despite substantial revisions and improvements by the authors in response to the reviewers' concerns, the paper still falls short in several critical aspects. While the revised results show improved performance and the method's efficacy in a broader range of applications, there are lingering issues regarding the scalability, general applicability, and universal capability of the CoI method across diverse domains. Additionally, concerns about the clarity of writing and the completeness of the methodology, though partially addressed, need further refinement. In light of these remaining issues, despite recognizing the improvements and the innovative aspects of the work, the paper does not fully meet the necessary standards for acceptance. Therefore, the recommendation is to reject the submission in its current form, with encouragement for the authors to further develop and refine their approach for future submission.

**Justification For Why Not Higher Score:**

Same as above.

**Justification For Why Not Lower Score:**

N/A

---

### Decision · Program_Chairs · 2024-01-16

Reject